# Modulating the activities of chloroplasts and mitochondria promotes adenosine triphosphate production and plant growth

Chia P. Voon[1], Yee-Song Law[1], Xiaoqian Guan[1], Shey-Li Lim[1], Zhou Xu[1], Wing-Tung Chu[1], Renshan Zhang[1], Feng Sun[1], Mathias Labs[2], Dario Leister[2], Mathias Pribil[3], Marie Hronková[4], Jiří Kubásek[4], Yong Cui[5,6], Liwen Jiang[5,6], Michito Tsuyama[7], Per Gardeström[8], Mikko Tikkanen[9] and Boon L. Lim[1,5,6]

[1]School of Biological Sciences, The University of Hong Kong, Pokfulam, China; [2]Plant Molecular Biology, Department of Biology, Ludwig-Maximilians-University Munich (LMU), Munich, Germany; [3]Copenhagen Plant Science Centre, Department of Plant and Environmental Sciences, University of Copenhagen, Copenhagen, Denmark; [4]Faculty of Science, University of South Bohemia, Ceske Budejovice, Czech Republic; [5]School of Life Sciences, Centre for Cell and Developmental Biology, The Chinese University of Hong Kong, Shatin, China; [6]State Key Laboratory of Agrobiotechnology, The Chinese University of Hong Kong, Shatin, China; [7]Department of Agriculture, Kyushu University, Fukuoka, Japan; [8]Umeå Plant Science Centre, Department of Plant Physiology, Umeå University, Umeå, Sweden; [9]Molecular Plant Biology, Department of Life Technologies, University of Turku, Turku, Finland

## Original Research Article

**Keywords:**
ATP; AtPAP2; chloroplasts; mitochondria; NADPH; photosynthesis.

**Author for correspondence:**
B. L. Lim, E-mail: bllim@hku.hk
Chia P. Voon, Yee-Song Law and Xiaoqian Guan contributed equally to this work.
Mathias Labs is currently with KWS SAAT SE, Gateway Research Center, St. Louis, Missouri, USA.

## Abstract

Efficient photosynthesis requires a balance of ATP and NADPH production/consumption in chloroplasts, and the exportation of reducing equivalents from chloroplasts is important for balancing stromal ATP/NADPH ratio. Here, we showed that the overexpression of purple acid phosphatase 2 on the outer membranes of chloroplasts and mitochondria can streamline the production and consumption of reducing equivalents in these two organelles, respectively. A higher capacity of consumption of reducing equivalents in mitochondria can indirectly help chloroplasts to balance the ATP/NADPH ratio in stroma and recycle NADP+, the electron acceptors of the linear electron flow (LEF). A higher rate of ATP and NADPH production from the LEF, a higher capacity of carbon fixation by the Calvin–Benson–Bassham (CBB) cycle and a greater consumption of NADH in mitochondria enhance photosynthesis in the chloroplasts, ATP production in the mitochondria and sucrose synthesis in the cytosol and eventually boost plant growth and seed yields in the overexpression lines.

## 1. Introduction

In plant cells, chloroplasts convert light energy into chemical energy, and mitochondria consume the chemical energy to produce ATP. The optimal carbon fixation and plant growth require these two energy-transforming organelles to perform strictly coordinated actions. Both organelles utilise protein complexes to construct electron transport chains (ETCs), which are responsible for the formation (chloroplasts) or consumption (mitochondria) of reducing equivalents, translocation of protons and the build-up of the proton gradient as a driving force for ATP synthesis. The cooperation between the chloroplasts and mitochondria, which was invented over 1.5 billion years ago, is nowadays much more complicated than the relationship between a supplier and a consumer in modern plant cells (Dutilleul et al., 2003; Noguchi & Yoshida, 2008). In chloroplasts, the linear electron flow (LEF) generates ATP/NADPH at a ratio of approximately 1.28, and the fixation of carbon dioxide consumes ATP/NADPH at a ratio of 1.5 (Allen, 2003; Foyer et al., 2012). Therefore, the photosynthetic efficiency requires the production and consumption of ATP and reductants at appropriate ratios in the chloroplasts, and this process is complicated by their fluxes across the chloroplast inner membrane. Our recent study showed that, in order to limit energy expenditure of chloroplasts in the dark, the

importation of cytosolic ATP into mature chloroplasts is negligible (Voon et al., 2018; Voon & Lim, 2019). Hence, during photosynthesis, the ATP/NADPH ratio can either be balanced by extra ATP production from the cyclic electron flow (CEF) or the export of excess reductants to the cytosol (Sato et al., 2019; Selinski & Scheibe, 2019). In addition, photorespiration also generates a large amount of NADH in mitochondria, and surplus reducing equivalents are exported to the cytosol through the malate–oxaloacetate (OAA) shuttle (Lim et al., 2020). Hence, surplus reducing equivalents generated during photosynthesis have to be stored as malate in the vacuole (Gerhardt et al., 1987). Light-dependent production of reducing equivalents in mitochondria is supported by the observation that illumination can cause a rapid pH change in the mitochondrial matrix, which was unseen when 3-(3,4-dichlorophenyl)-1,1-dimethylurea (DCMU) was applied (Voon et al., 2018). Furthermore, when the mitochondrial ETC was interfered by inhibitors, photosynthetic ATP production in stroma was also affected (Voon et al., 2018). Here, by studying the physiology of transgenic lines that overexpress *Arabidopsis thaliana* purple acid phosphatase 2 (AtPAP2), we showed that efficient cooperation of chloroplasts and mitochondria in optimising reductant production in chloroplasts and consumption in chloroplasts and mitochondria is important for enhancing photosynthesis and productivity.

AtPAP2 is anchored on the outer membranes of chloroplasts and mitochondria, and plays a role in protein import into these two organelles (Law et al., 2015; Sun et al., 2012a; Zhang et al., 2016). The overexpression of AtPAP2 in *Arabidopsis thaliana* resulted in earlier bolting (Supplementary Figure S1 and Supplementary Movie S1), a higher seed yield (+40–50%) and higher leaf sugar and ATP levels (Liang et al., 2015; Sun et al., 2013; Sun et al., 2012b). The overexpression of AtPAP2 also promotes plant growth and seed yield of the biofuel crop *Camelina sativa* (Zhang et al., 2012). Similarly to Toc33/34 and Tom20s, AtPAP2 is anchored onto the outer membranes of chloroplasts and mitochondria via its hydrophobic *C*-terminal motif (Sun et al., 2012a). AtPAP2 interacts with the precursor of the small subunit of RuBisCO (pSSU; Zhang et al., 2016) and the presequences of a number of multiple organellar RNA editing factor (pMORF) proteins (Law et al., 2015) and plays a role in their import into chloroplasts (Zhang et al., 2016) and mitochondria (Law et al., 2015), respectively. Here, we examined how AtPAP2 overexpression affects the physiology of chloroplasts and mitochondria and how these two energy-generating organelles orchestrate to produce more sugars and ATP in leaf cells. Our data suggest that the efficiency of photosynthesis is dependent on the activities of the mitochondria. Surplus reducing equivalents generated from the LEF have to be exported and dissipated, and mitochondria that more actively dissipate the reducing equivalents can supply more ATP to the cytosol, thereby simultaneously relieving the pressure of over-reduction of ETC in the chloroplast (Scheibe et al., 2005). This streamlined cooperation enables a higher efficacy in carbon fixation, sucrose synthesis and ATP production in leaf cells.

## 2. Methods

### 2.1. Plant growth conditions

The wild-type (WT) *Arabidopsis thaliana* ecotype Columbia-o (WT), the *Arabidopsis thaliana* AtPAP2-overexpressing (OE) lines (OE7 and OE21) and the *pap2* T-DNA insertion mutant (Salk_013567; Sun et al., 2012b) were used in this study. *Arabidopsis* seeds were plated for 10 days on Murashige and Skoog (MS) medium, which was supplemented with 2% (w/v) sucrose. Seedlings of similar sizes were transferred to soil under growth chamber conditions (120–150 μmol m$^{-2}$ s$^{-1}$, 16-h photoperiod, 22°C).

### 2.2. Chlorophyll a fluorescence measurement

The chlorophyll *a* fluorescence levels of the *Arabidopsis* leaves (20 days old) were monitored with IMAGING-PAM M-Series Maxi Version (Heinz Walz GmbH, Effeltrich, Germany). The plants were dark-adapted for at least 1 hr before the measurements. After the maximum and initial fluorescence ($F_m$ and $F_0$, respectively) were determined with a delay of 40 s, the plants were illuminated using the following light intensities: 0, 81, 145, 186, 281, 335, 461, 701 and 926 μmol photon m$^{-2}$ s$^{-1}$. The duration of the illumination of each light intensity was 3 min. A saturation pulse (800 ms, 2700 μmol photon m$^{-2}$ s$^{-1}$) was applied at the end of the 3-min illumination. For each light intensity, the photosynthesis parameters were calculated using the formulas: Y(II) was calculated as ($F_m$' − $F$) / $F_m$'; qP was calculated as ($F_m$' − $F$) / ($F_m$' − $F_0$'); Non-photochemical quenching (NPQ) was calculated as ($F_m$ − $F_m$') / $F_m$'; electron transport rate (ETR) was calculated as Y(II) × light intensity × 0.5 × absorptivity. $F_m$' was the maximal fluorescence of light-illuminated plant. $F$ was the current fluorescence yield. $F_0$' was estimated by $F_0$ / ($F_v$ / $F_m$ + $F_0$ / $F_m$') (Oxborough & Baker, 1997).

### 2.3. Measurements of P700 oxidation

P700 oxidation [Y(ND) = P / PM] was recorded with Dual-PAM-100 (Heinz Walz GmbH). P700 was determined by the difference of 875 and 830-nm measuring lights (Klughammer & Schreiber, 2008). Three-week-old plants were illuminated using the following light intensities: 0, 25, 50, 125, 450 and 150 μmol photon m$^{-2}$ s$^{-1}$. The duration of the illumination of each light intensity was 5 min, and saturating pulse (800 ms, 8,000 μmol photon m$^{-2}$ s$^{-1}$) was applied in 1-min intervals.

### 2.4. Measurements of CO$_2$ assimilation

The carbon dioxide assimilation in the 20-day-old plants was measured using the portable photosynthesis system LI-6400XT (LiCor, Lincoln, Nebraska, USA) and whole plant *Arabidopsis* chamber 6400-17. The plants were cultivated hydroponically in four-times diluted Hoagland nutrient solution (plants were fixed in 'Grodan'-mineral wool cubes) under growth chamber conditions (irradiation 120–150 μmol m$^{-2}$ s$^{-1}$, 16/8-h day/night photoperiod, temperature 22°C). The plants were transferred immediately before the measurement into conical pots dedicated to the 6400-17 chamber, and the boundary between the substrate (roots) and rosette was established using aluminium foil to minimise nonleaves transpiration. The light response of CO$_2$ assimilation was measured under ambient CO$_2$ concentration 400 μmol mol$^{-1}$ and PAR levels 2,000, 1,000, 500, 250, 120, 60, 30, 15 and 0 μmol photon m$^{-2}$ s$^{-1}$. Stable temperature at 25°C and the relative air humidity between 50 and 70% were kept in the chamber during measurement.

### 2.5. Measurements of the postillumination chlorophyll fluorescence transient

The modulated chlorophyll fluorescence was measured using a PAM 101 fluorometer (Heinz Walz GmbH). Before the minimal fluorescence ($F_0$) determination, the leaves were kept in darkness for at least 1 hr. The $F_0$ was induced by a red-modulated

measuring light with a photon flux density of approximately $0.2 \ \mu mol$ photon $m^{-2} \ s^{-1}$. The postillumination Chl fluorescence increase is attributed to the back flow of electrons to the PQ pool from NADPH in the stroma via the CEF around photosystem I (PSI), which depends on the NDH complex (Gotoh et al., 2010; Yamamoto et al., 2011).

### 2.6. Enzyme assays of the CBB cycle and TCA cycle

The chloroplast proteins were used to assay the fructose 1,6-bisphosphate aldolase (FBA) (Schaeffer et al., 1997) and $NADP^+$-glyceraldehyde-3-phosphate dehydrogenase (GAPDH) activities. Enzyme assays of the Tricarboxylic acid cycle (TCA) cycle were performed as described previously (Huang et al., 2015). The activity of the mitochondrial proteins (1–25 μg protein) were assayed spectrophotometrically using a Multiskan™ GO Microplate Spectrophotometer (Thermo Scientific, Waltham, Massachusetts, USA).

### 2.7. Measurements of oxygen consumption rate

The respiration rate of 21-day-old fully expanded leaf was measured with a Clark-type electrode (Oxygraph Plus, Hansatech, King's Lynn, United Kingdom) at room temperature in the dark. Leaves were infiltrated for 5 min in half strength MS medium and incubated in the dark for 10 min before measurement. The respiration rate was measured in the first 10 min. At the 10[th] minute and the 20[th] minute, 10-mM salicylhydroxamic acid (SHAM) and 4-mM potassium cyanide (KCN) were then added. AOX pathway respiration rate was defined as the sensitivity of $O_2$ uptake to 10-mM SHAM, and the COX pathway respiration rate was defined as the sensitivity of $O_2$ uptake to 4-mM KCN in the presence of 10-mM SHAM. The total respiration rate, alternative oxidase (AOX) respiration rate and cytochrome oxidase (COX) respiration rate were calculated as nanomoles of $O_2$ per minute per gram of fresh weight as described previously (Zhang et al., 2017b). Mitochondria were isolated (Law et al., 2015), and their integrity (Jacoby et al., 2015) was checked as described previously. The respiration rate of isolated mitochondria was measured at $25°C$ in a chamber containing 1 ml of the respiration buffer (0.3-M Suc, 5-mM $KH_2PO_4$, 10-mM TES, 10-mM NaCl, 2-mM $MgSO_4$ and 0.1% [w/v] BSA; pH 6.8 or pH 7.2). The capacity of complex I, internal and external NADH dehydrogenases and complex II were measured in the presence of deamino-NADH (1 mM), malate (10 mM), glutamate (10 mM) and NADH (1 mM) with or without ADP (100 μM) and/or rotenone (5 μM) as previously described (Jacoby et al., 2015; Meyer et al., 2009).

## 3. Results

### 3.1. Overexpression of AtPAP2 changes the composition of the thylakoid membrane

Using transmission electron microscopy (TEM), alterations in the ultrastructure of the thylakoid membrane of mesophyll chloroplasts were observed in the OE lines (Figure 1). The average diameters of the grana stacks were approximately 0.43 μm in all four lines, which is consistent with a previous report (Armbruster et al., 2013). Conversely, the average heights of the grana stack in the OE7 and OE21 chloroplasts were smaller than the average heights of the grana stack in the wild-type (WT) and *pap2* lines (Figure 1). The decreased stacking may be due to the decreased amount of photosystem II (PSII; Figure 2). An analysis of the photosynthetic pigments revealed that chlorophyll a, chlorophyll b, lutein and

$\beta$-carotene were reduced in the two OE lines. The reduced chlorophyll content in the OE lines correlates with the reduction in the grana stacking. The level of violaxanthin was, however, unaltered, suggesting that the xanthophyll cycle in the OE plants was suppressed to minimise the thermal loss of the excitation energy (Supplementary Table S1).

### 3.2. Overexpression of AtPAP2 changes the compositions of the photosystems and respiratory chain

Two-dimensional blue native/polyacrylamide gel electrophoresis (BN/PAGE) difference gel electrophoresis (DIGE) revealed that the abundances of specific photosystem components were altered in the OE7 chloroplasts (Figure 2a). The protein levels of the PSII core proteins PsbC and PsbB were significantly reduced in OE7, while the abundances of the PSI core proteins PsaA, PsaB and PsaD were unaltered, implying that the PSI to PSII complex ratio is higher in the OE7 chloroplasts than that in the WT (Supplementary Table S2). The protein abundances of PsbO1 and RbcL (RuBisCO large subunit) were increased by at least threefold in the chloroplasts in the OE7 line. The ATP synthase complex subunits were significantly reduced in the OE7 line, including ATPA, ATPB and ATPC (Supplementary Table S2). In the mitochondria, the protein abundances of the ATP synthase subunits alpha, beta and gamma were also lower in OE7 (Figure 2b and Supplementary Table S3).

### 3.3. LEF, but not the CEF, was enhanced in the OE lines

To investigate the impact of the altered thylakoid composition on the photosynthetic performance, the photosynthetic electron transport in 20-day-old leaves was assessed using chlorophyll fluorescence analysis (Figure 3 and Supplementary Figure S2). Both AtPAP2 OE lines displayed an increased PSII quantum yield Y(II), PSII photochemical capacity (qP) and ETR compared to the WT (Figure 3a), which suggests that the OE lines exhibited a higher LEF efficiency. No difference in terms of absorptivity was observed between the WT and OE lines, implying that the altered photosynthesis parameters in the OE lines were not due to any changes in chlorophyll absorption capacity (Supplementary Figure S3). An analysis of the P700 redox state showed that the PSI in the OE lines was highly oxidised under low-light conditions (25–125 μmol photon $m^{-2} \ s^{-1}$), suggesting that in the OE lines, the number of electrons that are consumed from the PSI by downstream pathways is higher than the number of electrons that are supplied by PSII to PSI (Figure 3b). This phenomenon was not observed when the electron flow was larger at the higher light intensities. Although the ETR and P700 redox analysis suggested that the OE lines have a higher electron transfer capacity, the NAD(P)H dehydrogenase- and Fd-dependent CEF were not significantly changed in the OE lines according to the postillumination chlorophyll fluorescence transient and ruptured chloroplast assay, respectively (Supplementary Figure S4). In summary, the LEF, but not the CEF, was enhanced in the OE lines, and PSI played a key role in this enhancement by relaying more electrons to $NADP^+$ in the transfer chain.

### 3.4. Photosynthesis rate was enhanced in the chloroplasts of the AtPAP2 OE lines

To study the effect of the altered thylakoid composition on the photosynthesis performance, the photosynthetic parameters were characterised in 20-day-old leaves by an analysis of gas exchange. Both OE lines exhibited a higher light-saturated carbon fixation

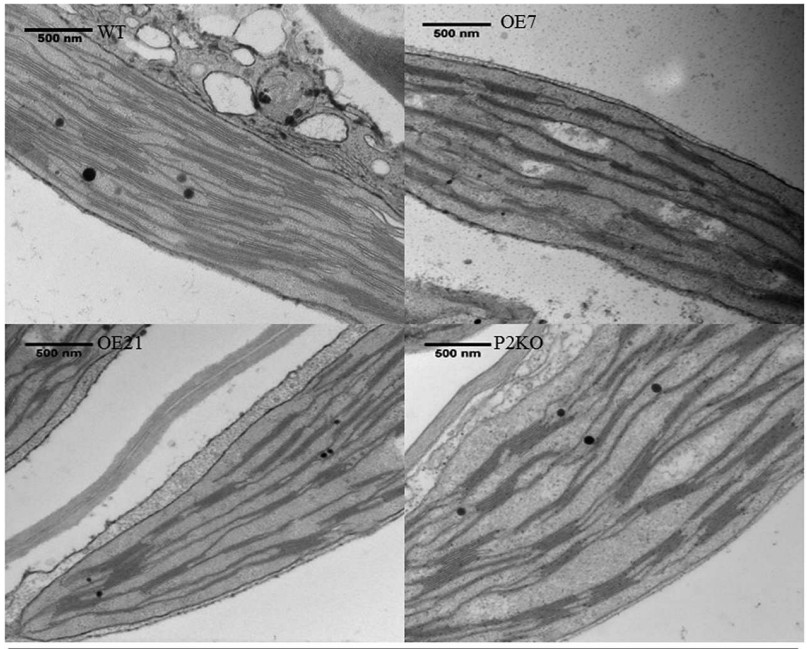

| Mean ± SEM | Diameter (nm) | Height (nm) |
|---|---|---|
| WT | 432 ± 19 [a] | 125 ± 8 [a] |
| *pap2* | 444 ± 11[a] | 102 ± 5[b] |
| OE7 | 431 ± 16 [a] | 78 ± 4[c] |
| OE21 | 430 ± 12 [a] | 79 ± 3[c] |

**Fig. 1.** Comparison of the thylakoid architecture between the wild-type (WT) and overexpressing (OE) plants. Transmission electron microscopic micrograph of ultrathin sections of 20-day-old leaves from the WT, OE (OE7 and OE21) and *Arabidopsis thaliana* purple acid phosphatase 2 knock-out (*pap2*) lines. Average value (*n* > 30) of the diameter (nanometre) and height (nanometre) of the thylakoid in the WT and OE lines were shown. Values marked by different letters in the same column are significantly different (*p* < 0.05) by Student's *t*-test. The diameter (nanometre) and height (nanometre) of WT chloroplasts isolated from 28-day-old leaves in another study (Armbruster et al., 2013) were 448 ± 16 and 113 ± 5 nm, respectively.

rate under ambient $CO_2$ concentration than the WT (Figure 3c). This increased rate correlates with the higher abundance of RbcL in the OE chloroplasts (Figure 2a). Assays of the CBB cycle enzymes showed that the capacities of fructose bisphosphate aldolase and GAPDH were increased by 30–40% in the chloroplasts in the OE7 line (Table 1).

### 3.5. The mitochondria in the AtPAP2 OE line had a higher capacity in dissipating reducing equivalents

The OE lines were shown to exhibit a higher total respiratory rate than the WT in the dark, with a significantly higher COX pathway respiratory rate (Supplementary Figure S5). Reductant equivalents can be exported from the chloroplasts in the form of malate or dihydroxyacetone phosphate (DHAP; Shameer et al., 2019), which can be used for sucrose synthesis in the cytosol or converted to pyruvate and fed into the TCA cycle. Enzyme assays of the TCA cycle suggested that the OE7 mitochondria had an increased capacity of mitochondrial $NAD^+$-dependent malate dehydrogenase ($NAD^+$-MDH) but not in the capacities of pyruvate dehydrogenase, citrate synthase or aconitase (Table 1), whereas the capacities of chloroplast $NADP^+$-dependent malate dehydrogenase ($NADP^+$-MDH), 2-oxoglutarate dehydrogenase and succinyl-CoA synthetase were increased by 1.26-fold, 1.15-fold, 2.86-fold and 10.3-fold in the OE7 line, respectively. Our recent studies showed that glycine decarboxylase generates a large amount of NADH in

mitochondria during photorespiration, which exceeds the NADH-dissipating capacity of the mitochondria. Consequently, the surplus NADH must be exported to the cytosol through the mitochondrial malate–OAA shuttle (Lim et al., 2020). The oxygraph studies showed that the electron-feeding capacities of complex II and internal NADH dehydrogenase, but not of complex I or external NADH dehydrogenase, were enhanced in the OE7 mitochondria (Supplementary Table S4). If the OE line had a higher capacity in dissipating NADH, less surplus reducing equivalents will be exported to the cytosol through the malate–OAA shuttle.

### 3.6. Higher mitochondrial activity is responsible for the higher cytosolic ATP in the OE line

Our previous study showed that the leaf in the AtPAP2 OE lines contained a significantly higher level of ATP and a higher ATP/NADPH ratio in both dark and light conditions (Liang et al., 2015). Here, we introduced a $MgATP^{2-}$ sensor (Imamura et al., 2009; Voon et al., 2018) into the chloroplasts and the cytosol and a pH sensor (Schwarzländer et al., 2011) to the mitochondrial matrix, and compared the responses of the OE7 and WT lines (Voon et al., 2018). Illumination led to an increased ATP concentration in the stroma (Supplementary Figure S6a) and alkalisation of the mitochondrial matrix in both lines (Supplementary Figure S6b). The extent of the alkalisation of the mitochondrial matrix was greater in the OE7 line than that in the WT, implying that under

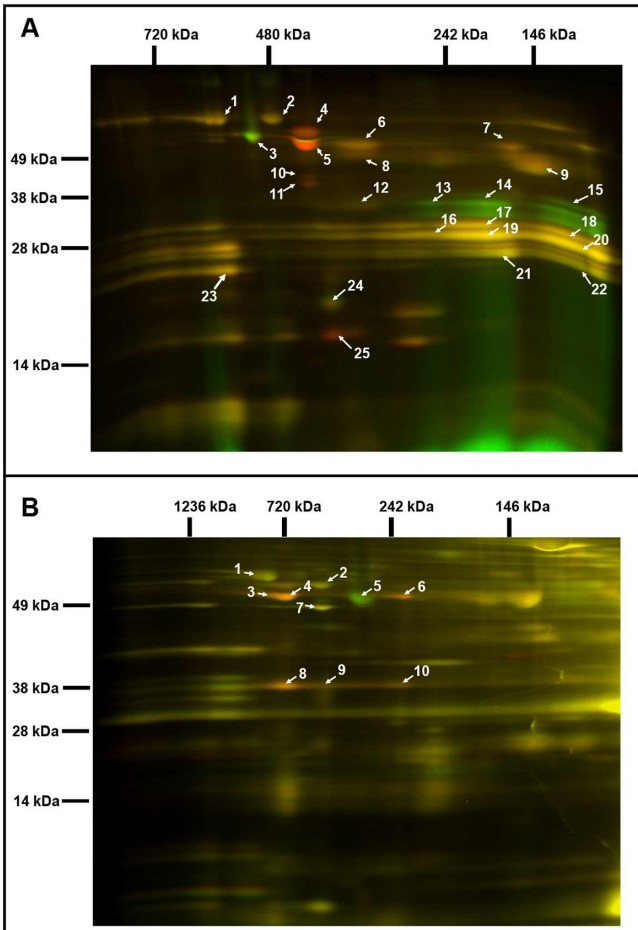

**Fig. 2.** Comparative analysis of the chloroplast and mitochondrial proteomes of wild-type (WT) and OE7 plants. (a) Chloroplasts. (b) Mitochondria. Proteins from the WT fraction were prelabelled with Cy3; proteins from the overexpressing fraction were prelabelled with Cy5. Combined protein fractions were separated by 2D BN/SDS-PAGE, and the protein visualisation was carried out by laser scanning at the respective wavelengths using the Typhoon laser scanner. On the resulting overlay image, Cy3 is represented by red, and Cy5 is represented by green. Proteins with a reduced abundance in the OE7 line are shown in red; proteins with an increased abundance in the OE7 line are shown in green; proteins of equal abundance in the two compared fractions are shown in yellow. Protein spots were extracted and identified by an MS/MS analysis. Identified proteins with the highest unused score and at least two unique peptides (95%) were labelled with the corresponding spot ID (Supplementary Tables S2 and S3). The representative gel images of three biological replicates are presented.

the same light intensity, the reducing power harvested by the chloroplasts may lead to a stronger proton translocation activity across the mitochondrial inner membrane (Supplementary Figure S6b). By contrast, the rate of increase in the ATP concentration in the stroma is lower in the OE7 line than that in the WT, which is likely due to a higher ATP consumption rate in the OE7 chloroplasts, because the $CO_2$ fixation rate (Figure 3c) and the capacities of certain CBB cycle enzymes are higher in the OE7 line (Table 1). We then examined the contribution of complex I, complex II and complex V in the OE7 mitochondria to cytosolic ATP. These inhibitors lower the cytosolic ATP in both lines (Supplementary Figure S6c). When complex I was inhibited by rotenone, the cytosolic ATP concentration was significantly higher in the OE7 line, suggesting that noncomplex I activities in the OE7 mitochondria, such as complex II and internal NADH dehydrogenase, can compensate more effectively than those in the

WT (Supplementary Table S4). This difference was not observed when complex II was inhibited by Thenoyltrifluoroacetone (TTFA). After 1 hr of incubation with oligomycin, cytosolic ATP in WT dropped to an undetectable level, whereas a higher level of cytosolic ATP was remained in OE7, suggesting that the intrinsic cytosolic ATP level was higher in OE7 than WT (Supplementary Figure S6c).

### 3.7. AtPAP2 selectively interacts with a number of chloroplast and mitochondrial proteins

A yeast two-hybrid (Y2H) library screening was carried out to identify other AtPAP2-interacting proteins. Forty nuclear-encoded AtPAP2-interacting proteins were identified; of these proteins, 32 and 3 proteins have been experimentally verified in chloroplasts and mitochondria, respectively (Supplementary Table S5). The other five AtPAP2-interacting proteins were predicted to be targeted to chloroplasts and/or mitochondria. AtPAP2 is exposed to the cytosolic side of the outer membrane of both organelles, and its role in the importation of organellar proteins may account for the enrichment of nuclear-encoded organellar proteins. A systematic Y2H assay was then carried out to examine the interactions between AtPAP2 and various nucleus-encoded photosystem proteins (Supplementary Figure S7a). AtPAP2 could specifically interact with Fd1, Fd2, PsaE2 (but not PsaE1), ferredoxin/thioredoxin reductase subunit A2 (FTRA2; but not FTRA1), FTRB and photosynthetic NDH subcomplexes L1, L2 and L3 (PsbQ-like1, PsbQ-like2 and PsbQ-like3). These interactions were verified by in vivo bimolecular fluorescence complementation (BiFC) assays (Supplementary Figures S7b and S8). Hence, AtPAP2 may play a role in the importation of these precursor proteins, particularly the proteins at the acceptor side of PSI, into chloroplasts.

### 4. Discussion

Maintenance of optimal photosynthetic efficiency requires the production and consumption of ATP and reductants in chloroplasts at appropriate ratios. Our recent study showed that the importation of cytosolic ATP into mature chloroplasts of *Arabidopsis thaliana* is negligible and therefore could not supply additional ATP to the CBB cycle (Voon et al., 2018). Hence, the export of surplus reducing equivalents to extrachloroplast compartments is essential for balancing the ATP/NADPH ratio (Voon & Lim, 2019). Under illumination, the chloroplast is the producer of reducing equivalents and ATP, and the mitochondria consume the excess reducing equivalents to supply ATP to the cytosol (Gardeström & Igamberdiev, 2016; Shameer et al., 2019; Voon et al., 2018). The results presented here provide an example on how the cooperation of chloroplasts and mitochondria boosts plant productivity.

### 4.1. Simultaneous alteration of chloroplast and mitochondrial activities leads to higher carbon fixation, higher leaf ATP but lower NADPH contents in the OE lines

Illumination greatly elevates the amount of NADPH and the NADPH/ATP ratio in WT leaves (Liang et al., 2016), which represents a high reduction state in plant cells. Our previous studies showed that in the middle-of-day, the leaf sucrose and ATP contents are higher in the OE lines than those in the WT line in 20-day-old plants, whereas the leaf NADPH contents, NADPH/NADP$^+$ and NADPH/ATP ratios are lower in the OE lines than that in the WT line (Liang et al., 2015). Here, our data

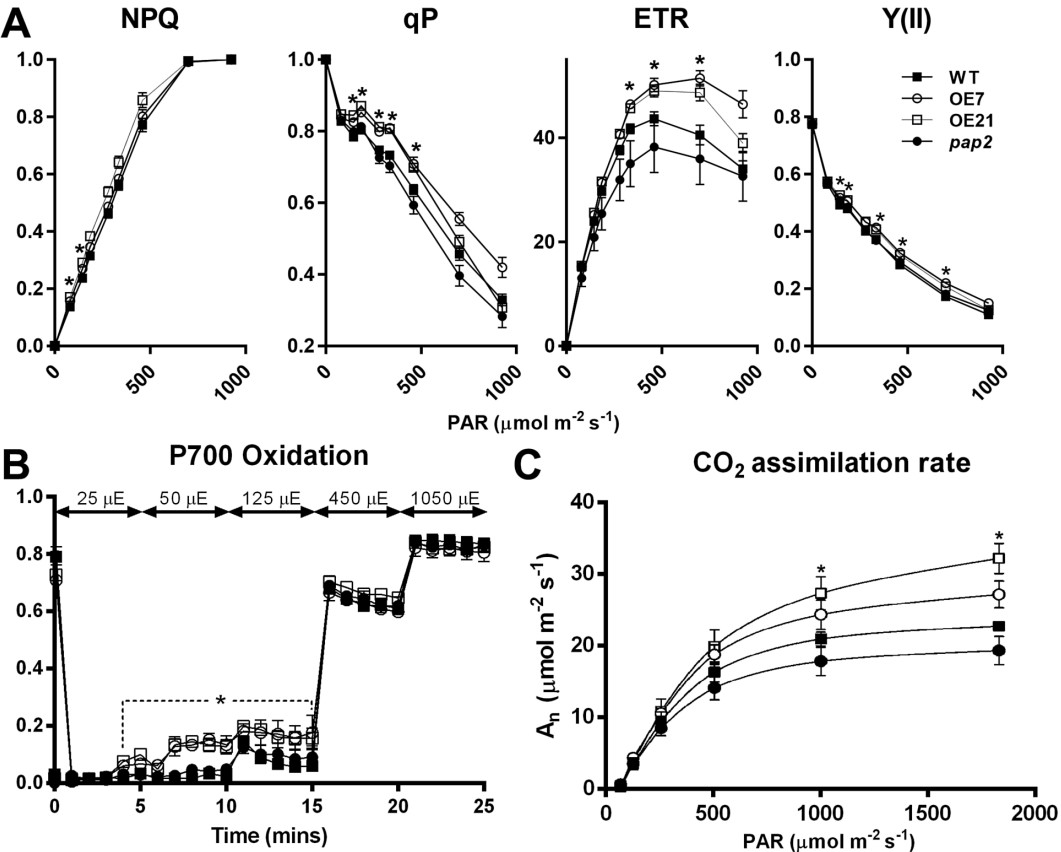

**Fig. 3.** In vivo analysis of the electron transport activity and $CO_2$ assimilation rate. (a) Light intensity-dependent NPQ, qP, ETR and Y(II). The 3-week-old plants were dark-acclimated for 1 hr before the measurement. Data are presented as the mean ± SE ($n > 10$ per line). (b) P700 oxidation. Data are presented as the mean ± SD ($n = 3$ per line). Light intensity was increased in a stepwise manner as stated in the graph. (c) Light response of $CO_2$ assimilation rates (An) under ambient $CO_2$ concentration. Data are presented as the mean ± SE ($n = 5$ per line). The asterisks indicate significant differences between the wild-type and both overexpressing lines by one-way ANOVA with post hoc Tukey HSD test ($p < 0.05$).

(Figure 4, Table 1 and Supplementary Table S4) indicated that the mitochondria of the OE line may have a higher capacity to consume extra reducing equivalents to generate ATP (Figure 4). This mechanism could account for the high ATP, high sucrose but low NADPH content in the leaves of the OE lines in the middle-of-day (Liang et al., 2015; Sun et al., 2012b). In addition, the RuBisCO content (Figure 2a) and the capacities of the CBB cycle enzymes FBA and GAPDH (Table 1) were higher in the OE chloroplasts. These could lead to a higher rate of $CO_2$ fixation (Figure 2c) and a rapid recycling of ADP, $NADP^+$ and ribulose-1,5-bisphosphate (RuBP) in the stroma. Hence, a faster recycling of RuBP and an enhanced output of DHAP from the OE chloroplasts could be achieved. This, together with a higher consumption rate of reductants by the mitochondria (Supplementary Table S4), might cause a lower reduction state in the stroma, thereby leading to an enhancement of the LEF in the OE lines. This hypothesis is supported by our data from the P700 redox analysis (Figure 3b). At low-light conditions (25–125 μmol photon $m^{-2}$ $s^{-1}$), in which the electron flow is smaller, PSI in the OE lines was highly oxidised, which was likely due to a high demand of electrons for the $NADP^+$ reduction or a higher PSI/PSII ratio (Figure 2a and Supplementary Table S2). This oxidised state was relieved under higher light conditions (>125 μmol photon $m^{-2}$ $s^{-1}$) when the electron flow was larger (Figure 3b).

### 4.2. The OE line has a higher PSI/PSII ratio, a higher LEF but indifferent CEF

In pea (*Pisum sativum*), the PSI/PSII ratios were adjusted under different light conditions as follows: sunlight-grown, 0.55; yellow light (preferentially excites PSII), 0.40; and red light (preferentially excites PSI), 0.91 (Chow et al., 1990). Compared to plants with a lower PSI/PSII ratio, plants with a higher PSI/PSII exhibit a higher rate of oxygen evolution under illumination intensities that exceed 200 μmol photon $m^{-2}$ $s^{-1}$ (Chow et al., 1990). We also observed a higher PSI/PSII ratio in the OE chloroplasts than that in the WT chloroplasts in the 2D-DIGE analysis (Figure 2a and Supplementary Table S2). The OE lines exhibited a higher ETR than the WT when the light intensity exceeded 200 μmol photon $m^{-2}$ $s^{-1}$ (Figure 3). A higher PSI/PSII ratio allows for a faster throughput of electrons from the photosynthetic ETC via the PSI acceptor side and results in a lower possibility of an over-reduction in ETC (Tikkanen et al., 2014). Interestingly, the OE lines had also less stacked thylakoid membranes compared to the WT (Figure 1). It is likely that the loose stacking increases the share of the grana margin domain that is composed of all components of the photosynthetic electron transfer chain in relation to the PSII-enriched grana stacks and PSI-enriched stroma lamellae. The importance of grana stacking for photosynthetic efficiency is not understood. Based on the results presented here, the loose stacking may be essential for the

**Table 1.** Enzyme activities of the CBB cycle and the TCA cycle in 20-day-old plants

| Enzyme | Source | WT | OE7 | Change |
|---|---|---|---|---|
| Fructose bisphosphate aldolase | Chloroplast | 92 + 8 | 123 + 14** | 1.34× |
| NADP+-GAPDH | Chloroplast | 97 + 11 | 138 + 4** | 1.42× |
| NADP+-MDH | Chloroplast | 328 + 20 | 413 + 40** | 1.26× |
| NADP+-MDH | Leaf | 26 + 4 | 28 + 4 | N.S. |
| NAD+-MDH | Leaf | 1,525 + 25 | 1,439 + 39 | N.S. |
| NAD+-MDH | Mitochondria | 4,657 + 206 | 5,354 + 46** | 1.15× |
| NAD+-ME | Mitochondria | 40 + 3 | 42 + 3 | N.S. |
| PDC | Mitochondria | 45 + 6 | 48 + 4 | N.S. |
| CS | Mitochondria | 84 + 12 | 90 + 12 | N.S. |
| ACN | Mitochondria | 314 + 18 | 322 + 18 | N.S. |
| NAD+-ICDH | Mitochondria | 43 + 3 | 63 + 8** | 1.47× |
| NADP+-ICDH | Leaf | 18.5 + 2.2 | 6.7 + 0.5** | 0.36× |
| 2-OGDH | Mitochondria | 1.4 + 0.6 | 4.0 + 0.4** | 2.86× |
| SDH | Mitochondria | 7.6 + 0.3 | 10.3 + 1.0** | 1.35× |
| Succinyl-CoA synthetase | Mitochondria | 29 + 1.4 | 300 + 63** | 10.3× |
| FUM | Mitochondria | 334 + 55 | 209 + 34** | 0.62× |

*Notes:* All enzyme units are presented as nmol/min/mg protein. Independent sample *t*-test was carried out. Significant differences between WT and OE7 are showed by asterisks. $n = 4$.
*Abbreviations:* 2-OGDH, 2-oxoglutarate dehydrogenase; ACN, aconitase; CS, citrate synthase; GAPDH, glyceraldehyde-3-phosphate dehydrogenase; FUM, fumarase; ICDH, isocitrate dehydrogenase; MDH, malate dehydrogenase; ME, malic enzyme; N.S., not significant; OE, overexpressing; PDC, pyruvate dehydrogenase complex; SDH, succinate dehydrogenase; WT, wild type. Values that are significantly differences between the two lines are marked by
* $p < 0.05$ and
** $p < 0.01$.

**Fig. 4.** A model on how efficient collaboration between chloroplasts and mitochondria promote ATP and sucrose production. ① and ② *Arabidopsis thaliana* purple acid phosphatase 2 on the outer membranes of chloroplasts and mitochondria promotes the import of certain proteins into these two organelles via the Toc or the Tom complexes. ③ Higher photosystem I/II ratio and higher LEF generate more NADPH and ATP at a ratio of 0.78, which are consumed at a ratio of 0.67 by the enhanced CBB enzymes in the overexpressing (OE) chloroplasts for $CO_2$ fixation. The surplus reducing equivalents are exported from the chloroplasts via the malate/oxaloacetate shuttle to recycle NADP+ as the electron acceptors of the LEF. ④ Higher reductant-dissipating activities of OE mitochondria reduce the needs for mitochondria to export reductants from photorespiration in the form of malate. ⑤ OE chloroplasts with enhanced rate of carbon fixation export more carbon skeletons to the cytosol. ⑥ OE mitochondria with higher reductant-dissipating activities generate more ATP through the respiratory electron transfer chain. ⑦ Higher ATP production in OE mitochondria and higher output of carbon skeletons from chloroplasts enhance sucrose synthesis in the cytosol. Red and blue lines indicate upregulated and downregulated pathways/metabolites in the OE lines, respectively.

efficient interactions among the components of the photosynthetic electron transfer chain.

Consistently with this hypothesis, the analysis of the P700 (PSI reaction centre) redox state revealed that the P700 in the OE lines was relatively oxidised compared to that in the WT when the elec-tron supply from PSII was limited (25–125 μmol photon m$^{-2}$ s$^{-1}$; Figure 3b). Y2H showed that AtPAP2 preferably interacts with the downstream components of PSI, including PsaE2, Fd1, Fd2, FTRA2 and FTRB (Supplementary Figure S7). Altogether, we hypothesise that the LEF rate in the OE lines was increased because of the

higher capacity of electron transfer in PSI. LEF generates 1.28 mole ATP per mole of NADPH, but a minimum of 1.5 mole of ATP per mole of NADPH is required for carbon fixation (Allen, 2003; Foyer et al., 2012). It is generally believed that CEF pathways are needed to supplement ATP and modulate the ATP/NADPH ratio to meet the demand of metabolism (Foyer et al., 2012; Ishikawa et al., 2016). However, our studies showed that an enhancement of CEF is not required to support a more robust CBB cycle in the OE line (Supplementary Figure S4).

### 4.3. The mitochondrial capacity in dissipating surplus reducing equivalents is important for photosynthesis

A recent study in diatom showed that the ATP generated from mitochondria can be transported to chloroplasts for carbon fixation (Bailleul et al., 2015), and it was believed that this also occurs in higher plants. However, our recent study showed that in the mesophyll of *Arabidopsis thaliana*, cytosolic ATP does not enter mature chloroplast to support carbon fixation (Voon et al., 2018). Altogether, because the NADPH/ATP demand for carbon dioxide fixation is approximately 0.67, and a 0.78 molecule of NADPH is generated from each ATP molecule in LEF (Allen, 2003; Foyer et al., 2012), one could expect that a surplus reducing power is generated from LEF under illumination. Other metabolisms in chloroplasts, such as transcription and translation, also consume ATP molecules; therefore, the surplus of reducing equivalents would be more evident. The surplus of reducing equivalents must be exported and dissipated to recycle $NADP^+$ for efficient photosynthesis.

The reducing power generated from LEF can be exported as DHAP and malate (Figure 4; Noctor & Foyer, 2000; Scheibe, 2004). While modelling predicted that only 5% or less of the reducing equivalents from LEF can be exported through the malate–OAA shuttle (Fridlyand et al., 1998), the higher capacity of the $NADP^+$-MDH in the OE chloroplasts might allow for the export of more reducing equivalents from the chloroplasts to the cytosol through the malate–OAA shuttle. Illumination promotes the association between mitochondria and chloroplasts and peroxisomes (Oikawa et al., 2015), and, therefore, the malate–OAA shuttle can be a very efficient energy and reductant transfer pathway between the chloroplasts, mitochondria and peroxisomes. Malate and OAA in the stroma and cytosol were estimated to have concentrations ranging from 1 to 3 and 0.025 to 0.098 mM, respectively (Heineke et al., 1991). This concentration ratio is maintained by the large positive value of free energy change, $\Delta G°'$, of malate dehydrogenation. This equilibrium can drive malate synthesis in the chloroplasts upon illumination, because the chloroplast $NADP^+$-dependent MDH is activated by light (Scheibe, 1987; Zhao et al., 2018). The reducing power, thus, can be readily channelled to the peroxisomes for hydroxypyruvate reduction through the malate–OAA shuttle. As more reductants are consumed by the more active OE mitochondria, less exportation of reductants from mitochondria is expected (Lim et al., 2020). Hence, more reductants from the chloroplasts can be dissipated indirectly (Figure 4). This is in line with our recent observation that the increase in stromal NADPH during illumination disappeared when photorespiration was absent (Abdel-Ghany, 2009; Lim et al., 2020).

### 4.4. The abundance of ATP synthases is not necessarily correlated with the cellular ATP levels

While the protein abundance of the ATP synthase subunits in the chloroplasts (Figure 2a and Supplementary Table S2) and mito-

chondria (Figure 2b and Supplementary Table S3) were significantly reduced in the OE7 line, the leaves of the OE lines contain higher ATP levels than the WT (Liang et al., 2015; Sun et al., 2013; Sun et al., 2012b). In Arabidopsis cotyledon, the cytosolic ATP concentration (>1.4 mM) is substantially higher than the stromal ATP concentration (~0.2 mM). Hence, the higher leaf ATP concentration of the OE line is likely to be contributed by the cytosolic ATP generated from the mitochondrial ATP synthase, which is supported by the cytosolic ATP sensor data (Supplementary Figure 6c) and oxygraph data (Supplementary Table S4). A lower abundance of ATP synthase does not contradict with a higher ATP content in the OE7 line. The output of ATP synthase is dependent on the magnitude of the proton motive force (Schmidt & Graber, 1987; Zubareva et al., 2020), and may also be affected by the fuel supply (NADH) to the mETC and ADP turnover and so forth. Despite of a lower abundance, the mitochondrial ATP synthase of the OE7 line may have a higher output than that of the WT.

### 4.5. Summary

In summary, the overexpression of AtPAP2 modulates the importation of selected nucleus-encoded proteins into chloroplasts (Zhang et al., 2016) and mitochondria (Law et al., 2015), which leads to stronger sinks of reductants, including higher activities in the CBB cycle and mitochondrial ETC, in both organelles. A higher output of reducing equivalents from the LEF of chloroplasts (Figure 3) and higher mitochondrial activities in utilising the reducing equivalents for ATP production, thus, contribute to the higher ATP content and ATP/NADPH ratio in the OE lines (Liang et al., 2015). The simultaneous activation of chloroplasts and mitochondria is required for the production of a surplus of ATP and sucrose; the transgenic lines that overexpress AtPAP2 solely in the chloroplasts grew similarly to the WT (unpublished data), and the transgenic lines that overexpress AtPAP2 solely in the mitochondria exhibited early senescence and a lower seed yield (Law et al., 2015). These results provide an example of how the efficient cooperation of chloroplasts and mitochondria can enhance ATP and sucrose production in leaf cells. While AtPAP2 can enhance yield and growth in *Arabidopsis thaliana*, it has negative impacts to plant survival. The OE lines are less resistant to drought and to *Pseudomonas syringae* infection (Zhang et al., 2017a). Therefore, while AtPAP2-like genes were evolved from green algae during evolution (Sun et al., 2012b), it may not be fully used by higher plants to promote growth in natural environment, as survival is more important than high yield. It may be more 'useful' for unicellular algae, as water is never a constraint, and algae need to control their growth rate and carbon level subject to the availability of P/N/Fe.

Over the past few decades, a controversy has emerged in the field regarding whether the shortfall of ATP for the CBB cycle is fulfilled by the CEF or the importation of ATP from the cytosol. Combining the findings of our recent study (Voon et al., 2018) and this study, we concluded that the efficient carbon fixation in the OE line is not dependent on an enhanced CEF or the importation of ATP from the cytosol to fulfil the shortfall of ATP generated from the LEF. Instead, an efficient carbon fixation is dependent on an enhanced LEF, enhanced capacities of CBB enzymes, the efficient export of surplus reducing equivalents from chloroplasts through the malate–OAA shuttle and a higher reductant-dissipating activity of mitochondria. The ability of mitochondria to dissipate reducing equivalents is important for relieving the built-up of surplus reducing power in the stroma, which, in excess, will limit the LEF due to an insufficient supply of $NADP^+$, the major electron

acceptor of the LEF. Our recent studies showed that the reducing equivalents from photorespiration are the major fuel for mitochondria in *Arabidopsis thaliana* (Lim et al., 2020). The optimal use of reducing equivalents by mitochondria not only generates more ATP for sucrose synthesis, but also reduces the export of reducing equivalents from the mitochondria and therefore the overall redox status of the photosynthetic cells. This provides a good example to illustrate the relationship between chloroplasts and mitochondria in bioenergetics.

## Acknowledgements

We thank Prof. Wah S. Chow, Dr. Boris Feniouk and Dr. Maurice Cheung for their comments on this manuscript. We thank Dr. Huang Shaobai and Dr. Richard Jacoby for training of YL in enzyme assays, oxygraph and 2D-DIGE. We thank Prof. Hans-Peter Braun and Dr. Holger Eubel for technical advice in the 2D BN/SDS-PAGE DIGE analysis.

**Financial support.** This project was supported by the Seed Funding Program for Basic Research (201311159043) of The University of Hong Kong, the General Research Fund (772012M) and the Area of Excellence Scheme (AoE/M-403/16) of the Hong Kong Research Grants Council and the Innovation and Technology Fund (Funding Support to State Key Laboratory of Agrobiotechnology) of the HKSAR, China. Access to LI-6400 was supported by the Czech research infrastructure for systems biology C4SYS (project no. LM2015055).

**Conflicts of interest.** AtPAP2 is the subject of US patent number 9,476,058.

**Authorship contributions.** CPV, YL and BLL wrote the manuscript with PG's input. CPV produced some of the transgenic sensor lines, performed the fluorescent sensor measurements and carried out the Maxi-Pam measurements and 2D BN-PAGE of the chloroplasts. YL carried out the 2D BN-PAGE of the mitochondria, the enzyme assays with WC and the mitochondrial oxygraph assays with ZX. XG screened the Y2H library and carried out the BiFC. SL carried out respiration, AOX and COX oxygraph assays. RZ and FS produced the overexpression lines. FS measured the chlorophyll content. ML, MP and DL carried out the chloroplast rupture assay, MTik carried out the P700 assay and MTsu carried out the postillumination chlorophyll fluorescence. JK and MH carried out the $CO_2$ exchange measurements. YC and LJ carried out the TEM analysis. BLL coordinated this study. All authors read and approved the manuscript.

**Data availability statement.** The authors declare that the main data supporting the findings of this study are available within the article and its Supplementary Information files. Extra data are available from the corresponding author upon request.

**Supplementary Materials.** To view supplementary material for this article, please visit https://dx.doi.org/10.1017/qpb.2021.7.

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
