## [Reviewer Report]

Dear Editor

We would like to submit a manuscript “Efficient cooperation of chloroplasts and mitochondria enhances ATP and sucrose production” to Quantitative Plant Biology as a research article. This manuscript is highly related to our recent publication "ATP compartmentation in plastids and cytosol of Arabidopsis thaliana revealed by fluorescent protein sensing" published in Oct 2018 and “In planta study of photosynthesis and photorespiration using NADPH and NADH/NAD+ fluorescent

protein sensors” in June 2020.

The first article illustrates how mature Arabidopsis chloroplasts maintain energy efficiency in the dark–by limiting ATP import from cytosol to mature chloroplast. It has been suggested that the shortfall of ATP in the Calvin-Benson-Bassham cycle could be compensated by the cyclic electron flow (CEF) or the importation of ATP from the cytosol. We reported in the first article that, unlike unicellular diatoms, mature Arabidopsis chloroplasts are unable to import ATP from the cytosol to supplement the ATP demand for CO2 fixation. Rather, the export of reducing equivalents is the key to maintain the optimal ATP/NADPH ratio required for photosynthesis. In the second article, we showed that photorespiration generates a large amount of NADH in mitochondria, and the surplus reducing equivalents is exported to the cytosol. Hence, surplus reducing equivalents are generated from both organelles during photosynthesis.

In the current manuscript, we showed that efficient carbon fixation in the AtPAP2 overexpression line is not dependent on an enhanced CEF to fulfil the shortage of ATP, but is dependent on an enhanced LEF and an efficient export of surplus reducing equivalents from the chloroplasts. Simultaneously, mitochondria play an important role in dissipating the surplus reducing equivalents from the chloroplasts, thereby regenerating NADP+ as electron acceptors for the LEF. By coordinating the activities of both chloroplasts and mitochondria, the AtPAP2-overexpressing lines can fix more carbon and produce more ATP and sucrose in their leaves. The OE plants thus grow faster and produce 50% more seeds (Supplemental Video S1). The current manuscript gives a real example to illustrate how the cooperation of chloroplasts and mitochondria can balance the demand and supply of ATP/NADPH during photosynthesis and optimize ATP and carbon fixation.

The content and authorship of this paper have been approved by all authors.

Yours sincerely

Boon Leong Lim

D.Phil (Oxon)

Associate Professor

School of Biological Sciences

University of Hong Kong

Hong Kong

---

## [Reviewer Report]

*Comments to Author*: In this manuscript, the authors examine underlying mechanisms of enhanced growth in PAP2-overexpressed Arabidopsis lines. They conducted physiological and biochemical experiments. They showed that the overexpression of PAP2 on the outer membranes of chloroplasts and mitochondria supports the oxidation of excess NADPH and balance the ATP/NADPH ratio, leading to enhance sucrose production and growth. Most of data were interesting, but the following flaws should be improved.

Major concerns

1) In L. 179, 209-210, the authors mentioned that the increased amount of RbsL leads to higher rate of CO2 assimilation in OE lines. If so, they should conduct A-Ci curve measurements using LI-6400XT, and compare Vcmax and Jmax values between WT and OE lines.

2) In L. 180-184 and Supplemental Tables S2 and S3, the authors showed the protein abundances of ATP synthase subunits in chloroplasts and mitochondria were lower in OE7 lines than WT. These results are contradictory to the description in L. 9-13.

3) Since in Fig. 3A, they measured CO2 assimilation rate of whole shoots, they can compare the respiratory CO2 efflux rate of whole shoots among lines.

4) In L. 165-166, 175-176 and Supplemental Table S2, the amounts of subunits of PSII and b6f complex are reduced in the OE7 lines, and these data are also contradictory to the description in L. 9-10, 189-191. How can the authors explain this contradiction? The increased amount of only PSI should relate to enhancement of CEF-PSI. Using their Dual-PAM data (L. 96-101), they can calculate and should show the data of electron flow rate of PSI and CEF-PSI.

5) In Fig. 2A, Fig. 2C, and Supplementary Figure S2 the authors measured photosynthesis of rosette leaves of whole shoots, but in Fig. 2B and Supplementary Figure S3 they measured photosynthesis of one leaf. Whole shoots include both immature young and senesced old leaves. These data should be carefully interpreted. What age of leaf is used for the measurements of Fig. 2B and Supplementary Figure S3?

6) In Table 2, if they would like to show the enhancement of respiratory ATP production in OE lines, they should show maximal COX and AOX activities using isolated mitochondria.

7) In Supplementary Table S1, the ratios of chlorophyll a to b are too low. Some chlorophyll a may be degraded to chlorophyll b. Also, these pigment contents should be expressed as leaf area basis because ETR and CO2 assimilation data are expressed as leaf area basis.

Minor concerns

1) In L. 88-89, the intensity of saturation pulse is low to reduce all QA. Did they check whether the intensity of saturation pulse is really saturated?

2) In L. 103-114, how did they measure rosette area of shoots?

3) In L. 125-132, did they isolate chloroplasts from rosette of whole shoots? And the isolation method should be described.

4) In L. 141-149 and Table 1, did they use whole leaf extracts or isolated organelle? Also they should the enzyme activities on the leaf area basis to compare the data of ETR and CO2 assimilation rate.

---

## [Reviewer Report]

*Comments to Author*: General comments

This manuscript by Voon et al describes the effects of overexpression of AtPAP2 on the chloroplasts and mitochondria, which finally lead the increase in the photosynthetic production. To the best of my knowledge, although there are lots of papers in which the authors are struggling against the improvement in photosynthetic production, most of them did not succeed to make it. So, the clear increase in photosynthetic production described in this paper should be published in the near future. To make this manuscript more solid, I’d like to ask the authors to reconsider about several points below.

Specific comments

1. Reconsideration about “Efficient cooperation of chloroplasts and mitochondria” is necessary

Probably, it is generally accepted nowadays that there is coordination between chloroplasts and mitochondria. So, if either chloroplasts or mitochondria are solely modified and then the other is affected, the authors could say the results is due to the cooperation. But, in the present study, the over-expressed AtPAP2 is localized to both chloroplasts and mitochondria. Thus, the authors should consider the possibility that the effects of the overexpression on two organella could be independent, not results of cooperation between the two. 

There is only one data that shows a modification on the cooperation between chloroplast and mitochondria in OE7 (Figure 4B) in which light-dependent alkalization was modified in the mutant plant. But I can’t find any other data or direct evidences in which such a modification in the response of mitochondria lead the enhancement of photosynthesis shown in Figure 3.

Between the three inhibitors in Figure 4C, rotenone and oligomycin can also affect photosynthesis. Therefore, the increased level of ATP could be explained only by the modification on chloroplast.

Therefore, the title “Efficient cooperation of chloroplasts and mitochondria”, the sentence “The results presented here provide an example on how chloroplast function is highly dependent on the mitochondria. (Line 283-284, page 11)” and relating discussion should be reconsidered.

Or, do I overlook or misunderstand important data which shows transport of metabolites between chloroplasts and mitochondria?

2. There is discrepancy between the parameters of photosynthetic light reaction and the CO_2_ assimilation

The level of oxidation of P700 is higher in OE lines at 50-125 uE, but CO_2_ assimilation is the same to WT under this light intensity (Figure 3). The authors explained that, in OE lines, there is an enhancement of electron transfer from PSI to NADP+, but the resultant NADPH should be utilized for CO_2_ assimilation. Therefore, CO_2_ assimilation also could be increased at the low light (50-125 uE), but not in the present study. Why does it happen? 

Similar discrepancy is also observed at high light. The CO_2_ assimilation in OE lines is higher at >1000 uE, but the oxidation of P700 is the same to WT under the high light conditions. Probably, “the same level of P700 oxidation” results in “the same level of NADPH production” between WT and OE lines if there are no differences in CEF. Not only the oxidation of P700, but other LTE parameters (qP, YII) in OE lines are also almost the same to WT in the high light. So, what does enhance the CO_2_ assimilation in OE lines under the high light?

I also found a strange explanation. In Figure 4A, the level of ATP in OE7 under the light is lower than WT. The authors explained as “due to a higher ATP consumption rate in the OE7 chloroplasts because the CO_2_ fixation rate (line 244-245)”. But, the light intensity is 296 uE (on the figure legend of Figure 4C), which caused no difference in CO_2_ assimilation between WT and OE7 (Figure 3C)…

Could you explain about why the light intensity which can enhances CO_2_ assimilation and that for light reaction-related parameters are distinct from each other?

3. Is the CEF activity in OE lines really the same to WT?

I’m not so familiar with the evaluation of CEF activity by looking at the small rise appeared immediately after turning off the actinic light, but in my eyes, there are some differences between WT and two OE lines (Supplemental Figure S3). In OE7, the initial rate of the rise looks faster than WT. On the other hand, its height in OE21 appears lower than WT. I’d like to ask the authors to explain precisely why they concluded the CEF activity in OE lines is the same to WT.

4. Please show the original kinetics of fluorescence from which the authors calculated the photosynthetic parameters

The authors show photosynthetic parameters on Figure 3 (NPQ, qP, ETR, YII) which are based on measurements of chlorophyll fluorescence. Please show the typical fluorescence kinetics for each line from which these parameters were calculated, because original kinetics sometimes contain very important information or something new even though it is not appeared on the calculated parameters. Even if the authors are currently not aware of them, the readers might find something important after publication in the future. So, I strongly recommend the authors to do it.

Minor points

- The discussion part should be divided into several sections, like as the Results. The current version is bit hard to read and understand.

---

## [Reviewer Report]

*Comments to Author*: Based on the comments of the reviewers and on my own evaluation, as a minimum requirement, I would like to invite the authors to address the remaining discrepancies (noted by both reviewers), which may only involve text changes.

---

## [Reviewer Report]

Dear Editor

Thank you for your email dated 3 Dec 2020, informing us that a revision is allowed. 

We thank the editor and the two referees for the encouraging and constructive comments on our manuscript. We carried out additional experiments and reported the new data in Supplementary Figure S5. We also provided a new Supplementary Figure S2 to present the kinetics data.

The manuscript has been carefully revised to address all points raised by the two reviewers and a point-by-point response to all comments is provided. We believe that the quality of the manuscript has been greatly improved and hope that the reviewers and editors will now find it acceptable for publication in Quantitative Plant Biology. All authors read the manuscript and approved both content and authorship. 

Thank you very much and we look forward to hearing from you. 

Yours sincerely,

Boon Leong

Corresponding author

---

## [Reviewer Report]

*Comments to Author*: In the revised version, the authors improved several concerned points, but major following flaws were not improved. I cannot recommend this manuscript can be acceptable for this journal.

1) Why did not the authors conduct A-Ci curve? They already used LI-6400-XT for photosynthesis measurements.

2) They should show the data of electron flow rates of both PSI and CEF-PSI using Dual-PAM.

3) Why did not they check whether the intensity of saturation pulse of IMAGING-PAM is really saturated?

4) I cannot agree with their thought that A lower abundance of ATP synthase subunits does not contradict with a higher ATP content in OE7 lines compared with WT. Also, I cannot agree with their thought that a lower abundance of PSII and b6f subunits does not necessarily contradict with a higher ETR.

---

## [Reviewer Report]

*Comments to Author*: This manuscript by Voon et al is the revised one describing the effects of overexpression of AtPAP2 on the chloroplasts and mitochondria. The authors made modification according to the reviewers’ comments, but I think that is not enough. This manuscript needs further edition.

1. The title MUST be changed. Although there are many data showing the effects of the overexpression of AtPAP2, it is possible that the alteration in the chloroplasts and that in the mitochondria could be independent, as I mentioned before. The authors seem to have got results from the transgenic plants in which AtPAP2 is overexpressed solely either chloroplast of mitochondria and their phenotype was different from the transgenic lines of present study (Lines 383-386). Based on this, the authors say the present results are due to the cooperation between the two kinds of organella. I could understand what they say, but considered objectively, this is a speculation. Do the authors have any data showing enhancement in transport or exchange of chemicals/ reductants/ proteins between the two organella in the transgenic plants? What is the “cooperation” the authors are thinking? Moreover, the authors’ reasoning for the “efficient cooperation of chloroplast and mitochondria” includes unpublished data, according to the response. I have to say again that there is NOT enough evidences showing efficient cooperation of chloroplast and mitochondria in the present work. Of course, the authors can discuss about these points, but I hate such a title leading a misunderstanding for readers.

Instead of the current title, I will suggest “Overexpression of AtPAP2 in both chloroplasts and mitochondria enhanced photosynthesis production in Arabidopsis”, or something like that, and strongly recommend putting Supplemental figure S1 as Figure 1. In any cases, the authors must change the title to the one showing more immediate and direct conclusion obtained from their present work, which should not include speculation based on unpublished results to avoid readers’ misunderstanding.

2. About the Supplemental figure S2. I would like to thank the authors for their trying to follow my advices, but these are not “fluorescence kinetics”. Fluorescence kinetics is a graph showing changes in fluorescence yield with the lapse of time. It should look like the first graph on the left of panel A in Supplemental figure S4. I’d like to ask the authors to show graphs of the fluorescence kinetics for each light intensity. All the graphs should include baseline before and after turning on the measuring light to show the level of Fo and Fo’ clearly.

---

## [Reviewer Report]

*Comments to Author*: The manuscript entitled "Efficient cooperation of chloroplasts and mitochondria enhances ATP and sucrose production" described in detail characterization of already established and characterize Arabidopsis plant lines overexpressing the AtPAP2 gene and T-DNA insertion lines. Here using various physiological measurements and 2D-based proteomics, the authors suggest that the observed boost in OE lines' plant growth is governed by the combination of higher photosynthetic efficiency and consumption of reducing equivalents in the mitochondria. All the experiments are well-conducted and portrayed, and essential controls are included.

Overall, the presented work represents an important contribution to the scientific community, especially to those engaging in developing approaches for improving plant performance.

I have only minor comments:

1. The lower level of ATP synthase in OE line contradict the author's claim regarding enhance photosynthesis activity in those lines- can this be explained?

2. Line 192-197. The authors discuss the P700 oxidation results as a proxy for increased NAPD+ reduction rates. However, highly oxidized P700 refers to the quenching state of PSI and not necessarily to the rate of electron transfer.

3. Why most of the data is presented as sup files??

4. Line 241-243, the authors discuss the observed light-induced alkalization as a proxy for proton translocation activity across the mitochondria inner membrane. However, a recent paper (https://www.biorxiv.org/content/10.1101/2020.10.31.363051v1) showed light-induced pH alteration also in the cytosol, which can also affect the pH in the mitochondria. Accordingly, the presented data not directly point to more robust proton translocation activity.

5. Line 285 As PAP2 expressed in both organelles, I would say that the data point for the cooperation of the chloroplast and mitochondria activities in boosting metabolism and not for the dependency of chloroplast function on mitochondrial metabolism.

6. Line 295 "OE mitochondria"?

7. It is not clear whether or not the protein expression analysis supports the author's hypothesis. A dedicated analysis that will examine the expression level of the 40 proteins shown to interact with PAP2 is essential. If the authors model is correct, a significant increase in the abundance of these proteins (compared to the all protein population) should be observed

---

## [Reviewer Report]

*Comments to Author*: Based on the comments of the reviewers and on my own evaluation, I would like to invite the authors, again, to carefully address the comments raised (noted by all reviewers), which might be addressed by appropriate text modifications.

---

## [Reviewer Report]

Dear Editor

We thank the three referees for their constructive comments on our submitted manuscript. Here we submitted a revision according to their comments. We followed the suggestion of reviewer 2 and changed the title of the manuscript to "Overexpression of AtPAP2 in both chloroplasts and mitochondria of Arabidopsis enhances plant growth". 

We provide a point-by-point response to all comments. We believe that our work has been strengthened by the revisions and hope that both reviewers and editor will now find it acceptable for publication in QPB. All authors read the manuscript and approved both content and authorship. We look forward to hearing from you. 

Yours sincerely,

Boon Leong LIM

---

## [Reviewer Report]

*Comments to Author*: In this version (QPB-20-0027.R2), the authors have revised the manuscript based on the comments raised by the reviewers. I would like to thank the authors for their efforts to answer all referees concerns which has substantially improved the quality of their work. Now, the manuscript in its current form is acceptable for publication in QPB.